# Development of Specific Molecular and Phenotypic Marker-Based Haploid Inducers in Rice

**Jian Wang** [1,†], **Huijing Yan** [1,2,†], **Xiaozhen Jiao** [1], **Jun Ren** [1], **Fengyue Hu** [1], **Huan Liang** [1], **Weihong Liang** [2,*] **and Chaolei Liu** [1,*]

1 State Key Laboratory of Rice Biology and Breeding, China National Rice Research Institute, Hangzhou 310006, China; 82101202227@caas.cn (J.W.); 2104283134@stu.htu.edu.cn (H.Y.); jiaoxiaozhen@caas.cn (X.J.); 82101191090@caas.cn (J.R.); 82101211093@caas.cn (F.H.); 82101215147@caas.cn (H.L.)
2 College of Life Sciences, Henan Normal University, Xinxiang 453007, China
* Correspondence: liangwh@htu.edu.cn (W.L.); liuchaolei@caas.cn (C.L.)
† These authors contributed equally to this work.

**Abstract:** Doubled haploid (DH) technology is an efficient strategy for producing completely homozygous lines for breeding programs. Mutations in the *MATRILINEAL* (*MTL*) phospholipase trigger intraspecific haploid induction in cereals. Although an in vivo haploid induction system based on *OsMTL*-edited plants has been established in rice (*Oryza sativa*), DH technology is still limited by other factors, such as haploid identification, which is one of the essential steps required for DH technology. In the study, we addressed this technical challenge by integrating specific molecular and phenotypic markers into rice haploid inducers. We first generated large fragment insertion or deletion mutations within the *OsMTL* gene and designed a pair of primers flanking the mutational sites to be used as the specific and universal molecular markers between wild-type and *Osmtl* plants. Next, we screened for hairy leaf as a single dominant trait and integrated it into specific molecular marker-based haploid inducers using the cross and self-cross method. When crossing cytoplasmic male sterile lines with these haploid inducers, we utilized the specific InDel marker and hairy leaf phenotypic marker to identify putative haploids (or double haploids). These putative haploids were further confirmed through ploidy and phenotypic analysis, demonstrating the high efficiency of haploid identification using these markers. The haploid induction rate (HIR) of the developed specific molecular and phenotypic marker-based haploid inducers ranged from 3.7% to 12.5%. We have achieved successful integration of distinct molecular and phenotypic markers into rice haploid inducers. Our advanced marker-based system has significantly enhanced the accuracy of haploid identification, thereby expediting the adoption of DH technology in rice breeding.

**Keywords:** rice; DH technology; *MTL*; haploid identification; marker

## 1. Introduction

Compared with traditional methods of inbred line development, doubled haploid (DH) technology is a more efficient strategy for producing completely homozygous lines [1]. Although both in vivo and in vitro methods are available to generate haploids, only in vivo intraspecific hybridization using haploid inducers provides a promising approach for large-scale DH line production [2]. In maize, a haploid induction line called Stock 6, which was firstly reported in 1959, can produce 2–3% maternal haploid seed during self-pollination or when outcrossed as a male [3]. Stock 6 has contributed to the commercialization of maize haploid breeding, as the haploid induction rate (HIR) has been increased to 7–15% by academic and commercial routine users [4]. Recently, the genetic basis of haploid induction has been identified as two main molecular players in maize: *MTL* is responsible for haploid induction [4–6], and domain membrane protein (*DMP*) enhances haploid induction [7]. *MTL* is conserved in cereals, and knockout of *OsMTL* in rice can be used as a haploid

inducer [8–10], which takes the first step towards developing an in vivo haploid induction system in rice.

DH technology comprises four distinct steps: haploid induction, haploid identification, genome doubling, and seed production from DH lines [1]. As an in vivo haploid induction system is newly emerging in rice, a number of issues should be addressed in each procedure [9]. Haploid identification, a key procedure in DH breeding technology, is highly limited by the efficiency of identifying haploids from diploids due to the low frequency of haploid induction [1]. Developing an efficient method for haploid identification will greatly increase DH technology efficiency and reduce downstream process costs. In maize, haploids can be efficiently identified by using genetic and phenotypic markers that are integrated in the haploid inducers, such as the *R1-nj* based anthocyanin marker [11], the high oil content marker [12], the red root marker [13], and the *ZmC1/ZmR2*-based anthocyanin marker [14]. However, efficient methods for distinguishing haploids from diploids are lacking in rice DH technology.

Haploid induction crossing between *mtl* haploid inducers and source germplasm generates diploids that contain both the male and female parents' genome, as well as maternal haploids that carry only the female parent genome [1]. Applying this principle, unique molecular markers can be integrated into haploid inducers to aid in haploid identification. To achieve this, we propose utilizing CRISPR/Cas9 technology to create large insertion/deletion (InDel) mutations within the *OsMTL* gene, thereby integrating specific and universal genetic markers into rice haploid inducers. Additionally, we conduct a meticulous review of germplasms in order to identify dominant and visually discernible phenotypic markers. Next, we intend to incorporate these markers into our molecular marker-based haploid inducers, thereby boosting our overall efficiency.

## 2. Materials and Methods

### 2.1. Plant Transformation

In this study, we utilized two *OsMTL* genome editing vectors from our previous research [9], namely pC1300-ACT:Cas9-sgRNA$^{Target1}$ and pC1300-ACT:Cas9-sgRNA$^{Target2}$, and introduced them into the typical *indica-japanica* hybrid rice 'YongYou1540' (YY1540). Following plasmid construction, the *Agrobacterium*-mediated transformation (strain EHA105) experiments were conducted by Hangzhou Biogle Co., Ltd. (Hangzhou, China).

### 2.2. Detection of Mutations

To detect mutations, fresh leaf tissue from transgenic plants (100 mg) was pulverized using a tissuelyser (Jingxin, Shanghai). The genomic DNA was extracted using the CTAB method. Amplification of the fragments flanking the two targeted sites was performed using KOD FX DNA polymerase (Toyobo, Japan) and genotyped via Hi-TOM technology [15]. The Hi-TOM primers utilized for amplification are listed in Supplementary Table S1.

### 2.3. Plant Growth Conditions

After being soaked in water at 37 °C for two days, the seeds were sown in a seedbed. After 25 days, the seedlings were transplanted to a transgenic field, with six plants per row. Plant materials were cultivated following the local (Hangzhou, China) standard agronomic practices during the summer growing season. In winter, the plants were grown in a greenhouse located at the China National Rice Research Institute (CNRRI) in Hangzhou, China, under conditions that maintained an average temperature of 30 °C during the day and 25 °C at night, with a 16 h light/8 h dark cycle and 75% relative humidity.

### 2.4. HIR Measurement

Investigation of HIR involved crossing ZhongguangA with haploid inducers to produce hybrids. Following harvesting, the seeds were germinated and cultured in the hydroponic nutrient solution as previously described by Liu et al. [16]. Subsequently, genomic DNA was extracted from each individual using the CTAB method. The 2 × Taq

Master Mix was used to amplify target sequences with InDel marker 1 and 2, and PCR products were analyzed by running them on 5% agarose gels. Putative haploids were then confirmed using a flow cytometer.

*2.5. Ploidy Analysis*

The material used for ploidy analysis comprised of approximately 2 cm$^2$ of fresh leaf tissue, which was prepared following a previously described method [17]. Subsequently, the ploidy of the sample was determined using the BD Accuri C6 flow cytometer, with laser illumination at 552 nm and a 610/20 nm filter.

## 3. Results

*3.1. Knockout of OsMTL in YongYou1540*

The rice *OsMTL* gene encodes a sperm-specific phospholipase [8] and has four exons and three introns (Figure 1). Previously, we designed and developed two CRISPR/Cas9 vectors targeting exon 1 and exon 4 of *OsMTL*, respectively [9] (Figure 1). The two vectors were introduced into YongYou1540 (YY1540) for transformation. In the T$_0$ generation, a total of 31 and 48 independent transgenic plants were obtained for target 1 and 2, respectively. We genotyped these lines and detected six heterozygous or homozygous large insertion/deletion mutants: line M1 (43 bp deletion) at the target 1 site, and line M2 (26 bp insertion), M3 (29 bp deletion), M4 (22 bp deletion), M5 (34 bp deletion), and M6 (13 bp deletion) at the target 2 site (Figure 2). These mutants were reproduced to the T$_1$ generation to generate homozygous mutants and sufficient seeds for subsequent studies.

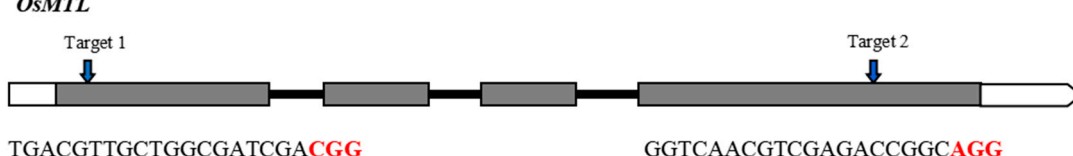

**Figure 1.** *OsMTL* gene structure and sgRNA target sites. Exons and introns are shown as boxes and lines, respectively. PAM motifs are highlighted in red.

**Target 1**

| WT | CGGCGTCGGCGCCGGGGCAGCGGGTGACGTTGCTGGCGATCGACGGCGGCGGCATCA | |
| --- | --- | --- |
| M1 | CGGCGTC----------------------------------------CGGCATCA | 43 bp deletion |

**Target 2**

| WT | GATGCTGGCGCAGCGGGTGTCGAGGGTCAACGTCGAGACCGGCAGGTACGTCGAGGT | |
| --- | --- | --- |
| M2 | CGAGGGTCAACGTCGAGATGCCCGGCGCCGGTACGTCGAGGTGCACGGCAGGTACGT | 26 bp insertion/SNP |
| M3 | GATGCTGGCGCAGC-------------------------AGGTACGTCGAGGT | 29 bp deletion |
| M4 | GATGCTGGCGCAGCGGG--------------------CGGCAGGTACGTCGAGGT | 22 bp deletion |
| M5 | GATGCTGGC------------------------------AGGTACGTCGAGGT | 34 bp deletion |
| M6 | GATGCTGGCGCAGCGGGTGTCGAGGGT------------GGCAGGTACGTCGAGGT | 13 bp deletion |

**Figure 2.** Generation of large insertion/deletion mutations in the *OsMTL* gene. The hybrid rice YongYou1540 was used for genetic transformation. Blue and red letters indicate the target and PAM sequences, respectively. The dashed lines represent nucleotide deletions. Insertions and SNPs are shaded in yellow, and the size of the deletion or insertion is shown on the right.

*3.2. Integration of Specific Molecular Markers into mtl Mutants*

The maternal haploids produced by *mtl* haploid inducers exclusively harbor the genome of the female parent [4–6,8], while the undesired diploids contain genomes from both the male and female parents. Therefore, molecular markers developed based on sequence differences between male and female parents can be employed to identify haploids. However, the heterozygous and variable breeding materials to be made homozygous

limit the application of molecular markers in haploid identification. To integrate unique and universal molecular markers into the *mtl* haploid inducers, we designed two pairs of primers that flank targets 1 and 2. The PCR products were 132 bp and 100 bp with reference to the normal genome (Figure 3). The large insertion or deletion mutations in the *mtl* haploid inducers (Figure 2) allowed the designed primers to serve as unique and universal InDel markers between the wild-type (WT) and *mtl* mutants. We further tested the polymorphism of the two InDel markers and found they could be easily distinguished on a 5% agarose gel (Figure 4). Therefore, we successfully integrated specific molecular markers into *mtl* haploid inducers.

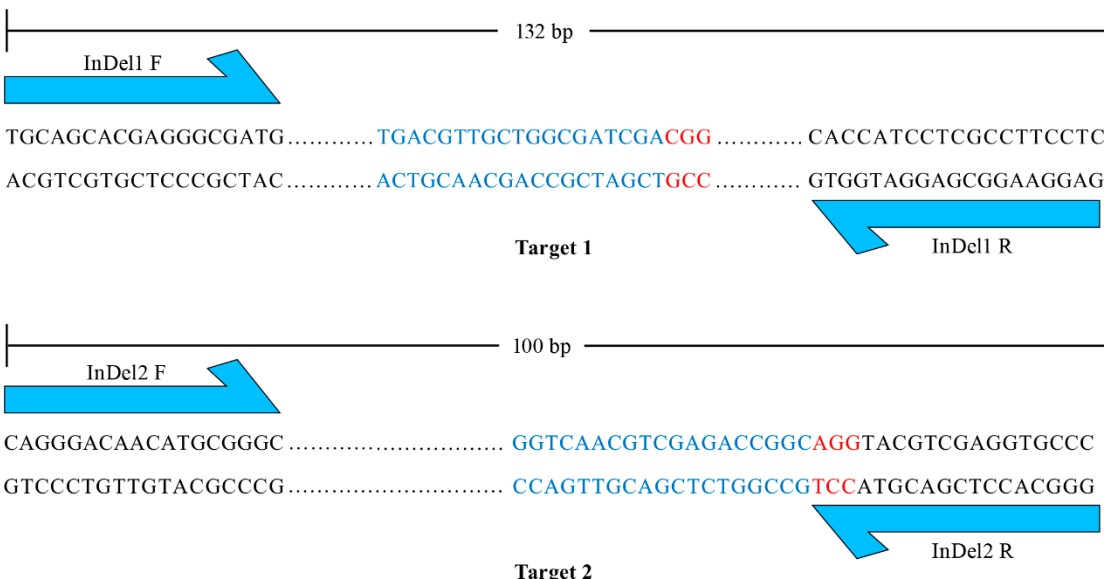

**Figure 3.** Design of specific molecular markers. The blue one-way arrows represent the sites of forward and reverse primers of the InDel markers. Targets and PAMs are indicated by blue and red letters, respectively. The dots represent the sequences between the primer and the target.

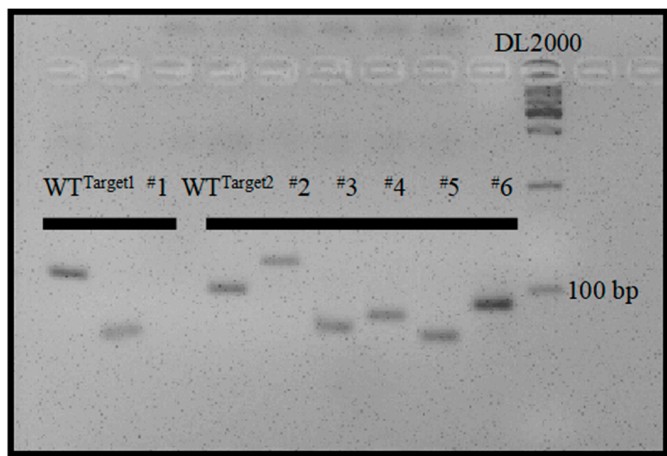

**Figure 4.** Polymorphism analysis of InDel1 and InDel2 markers in 5% agarose gel. The homozygous mutants of M1, M2, M3, M4, M5, and M6 from the $T_1$ generation were used for analysis.

### 3.3. Development of Specific Phenotypic Marker Haploid Inducers

Phenotypic markers, such as the *R1-nj* based anthocyanin marker, greatly improve the efficiency of haploid identification in maize DH technology [11]. To identify visually discernible phenotypic markers in rice, we conducted a meticulous review of germplasms and identified the NWFB10 variety, which shows hairy leaves at the seedling stage and

can be used as a phenotypic marker. The dense trichomes appear on the leaf surface of NWFB10 seedlings 10 days after germination and become increasingly prominent as the plant matures. To confirm that hairy leaf can be employed in haploid identification, we crossed the NWFB10 plant with the 93–11 variety (hairless leaf) and observed that all the $F_1$ offspring had hairy leaves (Figure 5a). In the segregating population of 108 $F_2$ plants, 76 plants displayed a hairy leaf phenotype similar to NWFB10, while 32 plants exhibited hairless leaves, fitting the typical segregation ratio of 3:1 ($\chi 2 = 1.23 < \chi 0.05 = 3.84$). These results indicate that hairy leaf is a single dominant trait and can be used as a phenotypic marker in haploid identification.

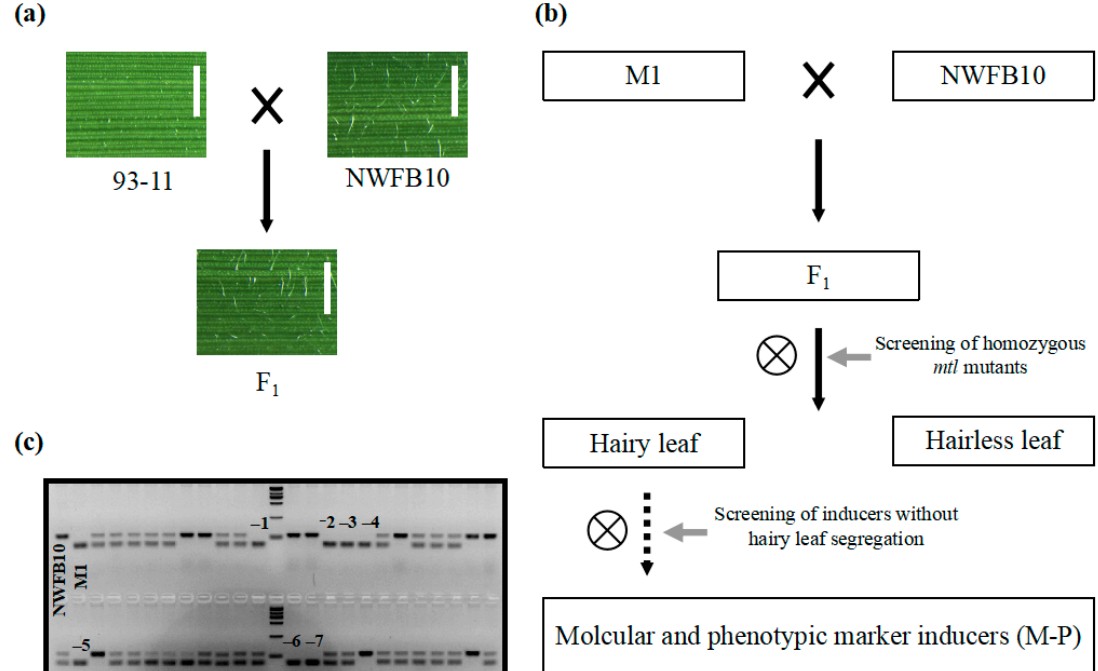

**Figure 5.** Development of hairy leaf marker-based haploid inducers. (**a**) Verification of the hairy leaf trait as a dominant characteristic. Bar = 1 mm. (**b**) Flow chart of the method used to develop hairy leaf marker haploid inducers. The solid and dotted arrows represent one generation and more than two generations, respectively. (**c**) Gel electrophoresis analysis to screen for homozygous *mtl* mutants. InDel1 marker was used for PCR amplification. The black numbers represent genotypes that are the same as the M1 haploid inducer. Marker = 2000 bp.

We then utilized the hairy leaf trait as a phenotypic marker to increase the efficiency of haploid identification in haploid inducers. Figure 5b displays a brief flow chart outlining the development of hairy leaf-based haploid inducers. Initially, we crossed the *mtl* haploid inducers with the NWFB10 variety and screened for lines exhibiting both the *mtl* mutation and the hairy leaf phenotype. Subsequently, we allowed the selected lines to self-fertilize and screened for lines with no hairless leaf phenotype in their progeny. We crossed the M1 haploid inducer with NWFB10 and identified seven *mtl* homozygous lines using the InDel1 marker in a segregating population of 46 $F_2$ plants (Figure 5c). We harvested and grew the seeds of the seven *mtl* mutants separately for phenotypic analysis. Three lines, namely M-P1, M-P2, and M-P3, lacked segregation for the hairless leaf phenotype in their progeny and were therefore identified as phenotypic marker-based haploid inducers with both InDel and hairy leaf markers.

### 3.4. Identification of Putative Haploids Using Molecular and Phenotypic Markers

To test the efficiency of haploid identification by using the specific InDel and phenotypic markers, we used the cytoplasm male sterility (CMS) line ZhongguangA to cross with these haploid inducers and obtained a set number of hybrid seeds.

We first evaluated the efficiency of utilizing specific molecular markers to identify haploids. We germinated the hybrid seeds, extracted the genomic DNA of each individual, and amplified the fragments using InDel1 or InDel2 markers. After amplification, we electrophoresed the PCR products in a 5% agarose gel. Due to the elimination of paternal chromosomes that occurred during haploid induction [18], maternal haploids or spontaneously doubling haploids only showed one band consistent with ZhongguangA at the target sites. On the other hand, heterozygous diploids contained both ZhongguangA and mutant fragments, giving rise to two bands (Figure 6). Overall, we identified 10, 17, 11, 12, 8, and 10 putative haploids or double haploids out of 100, 159, 116, 96, 137, and 94 hybrid seeds from the cross between ZhongguangA and haploid inducer M1, M2, M3, M4, M5, and M6, respectively (Table 1), with a haploid induction rate (HIR) of 10.0%, 10.7%, 9.5%, 12.5%, 5.8%, and 10.6%.

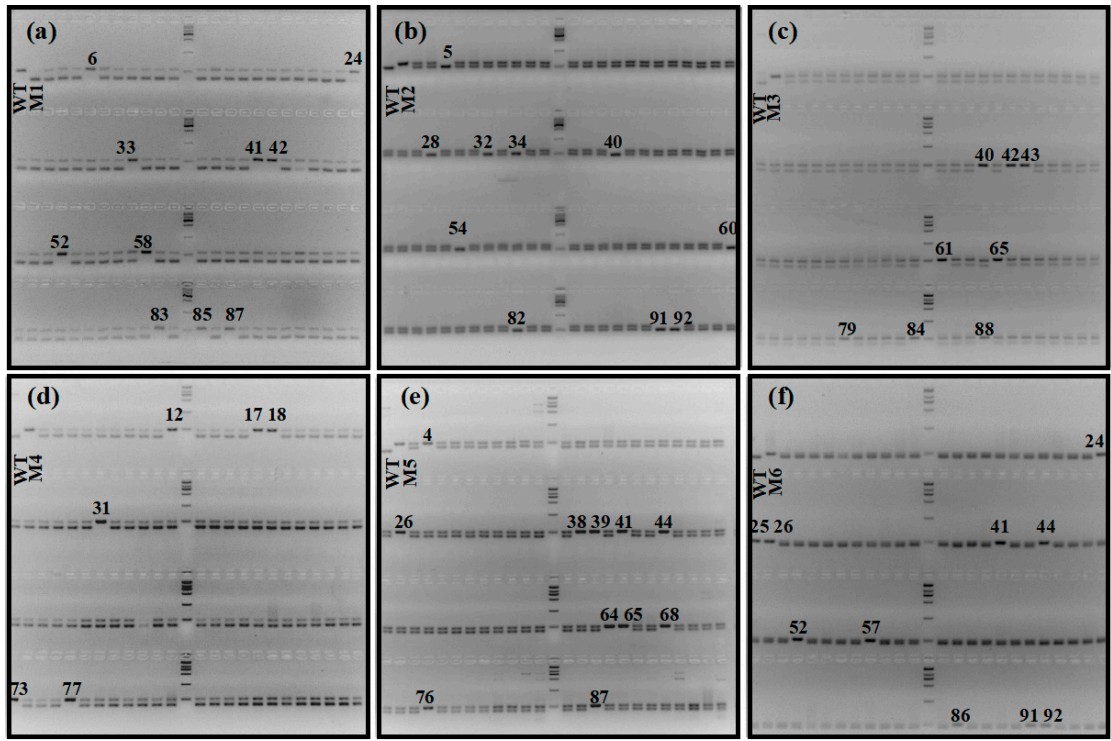

**Figure 6.** The represented gel electrophoresis analysis results of hybrid seedlings derived from crosses between cytoplasm male sterility line ZhongguangA and haploid inducers M1 (**a**), M2 (**b**), M3 (**c**), M4 (**d**), M5 (**e**), and M6 (**f**). Haploids or double haploids are marked with black numbers. WT represents the genome type of ZhongguangA. Marker = 2000 bp.

We tested the effectiveness of using hairy leaf phenotypic marker to identify haploids. We analyzed 156, 261, and 188 hybrid seeds obtained from the cross between ZhongguangA and the haploid inducers M-P1, M-P2, and M-P3. Among the seeds, 8, 11, and 7 lines exhibited the hairless leaf phenotype, while the rest showed the hairy leaf phenotype (Figure 7a). We considered the hairless leaf individuals as potential haploids or double haploids and verified this by using the InDel1 marker (Figure 7b). Out of these individuals, 6, 10, and 5 showed one band consistent with ZhongguangA. Therefore, the haploid induction rate (HIR) for M-P1, M-P2, and M-P3 was found to be 5.1%, 4.2%, and 3.7%, respectively. Although a few hairless leaf individuals exhibited two bands, which may be

due to the underdevelopment of trichome formation in the early seedling stage, using the hairy leaf marker remains an efficient method for haploid identification.

**Table 1.** Haploid induction rate (HIR) of the developed molecular and phenotypic markers based haploid inducers.

| Haploid Inducers | | Putative H/DH | H/DH | Progeny | HIR (%) |
|---|---|---|---|---|---|
| Molecular marker | M1 | 10 [a] | 10 | 100 | 10 |
| | M2 | 17 [a] | 17 | 159 | 10.7 |
| | M3 | 11 [a] | 11 | 116 | 9.5 |
| | M4 | 12 [a] | 12 | 96 | 12.5 |
| | M5 | 8 [a] | 8 | 137 | 5.8 |
| | M6 | 10 [a] | 10 | 94 | 10.6 |
| Phenotypic marker | M-P1 | 6 [b] | 8 | 156 | 5.1 |
| | M-P2 | 11 [b] | 11 | 261 | 4.2 |
| | M-P3 | 5 [b] | 7 | 188 | 3.7 |

Note: [a] and [b] represent haploids or double haploids identified by the specific InDel marker or hairy leaf marker, respectively.

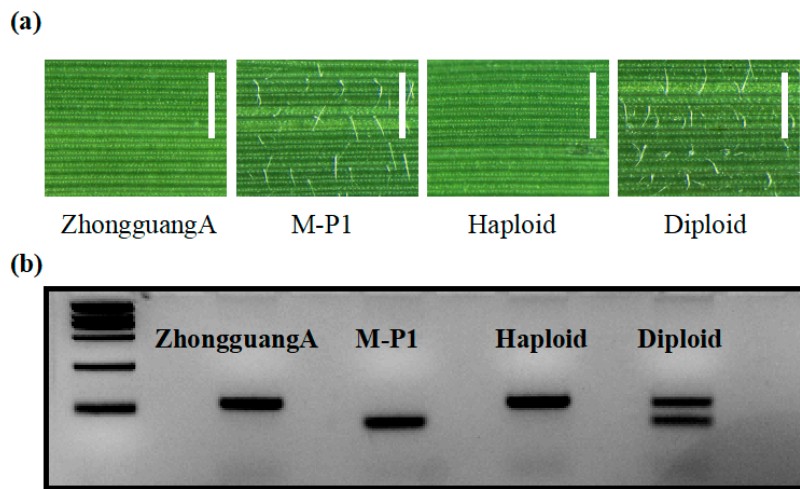

**Figure 7.** Identification of haploids using the hairy leaf marker. (**a**) Adaxial leaf epidermis morphology of ZhongguangA, M-P1, haploid, and diploid observed under light microscopy. Bar = 1 mm. (**b**) Polymorphism analysis of ZhongguangA, M-P1, haploid, and diploid was conducted using the InDel1 marker. The results were visualized in a 5% agarose gel. Marker = 2000 bp.

### 3.5. Verification of Haploids by Ploidy and Phenotypic Analysis

To verify the feasibility of haploid identification using specific molecular and phenotypic markers, we firstly determined the ploidy of putative haploids or doubled haploids (DHs) through flow cytometry. The ploidy of putative haploids 6, 24, and 33, which were harvested from the hybrid seeds of ZhongguangA and haploid inducer M1, was half of the heterozygous diploid used as the control (Figure 8a). This observation indicated the high accuracy of haploid identification through the specified markers. Following that, the haploids were transplanted to the field to analyze their phenotypic characteristics. At maturity, haploids exhibited a significant decrease in height compared to diploid plants, exhibited dwarf plant height, short spikelets, small glume size, and complete sterility (Figure 8b). The dwarf and small phenotypes of haploids were due to the half-reduced genome ploidy level, as previously reported [19].

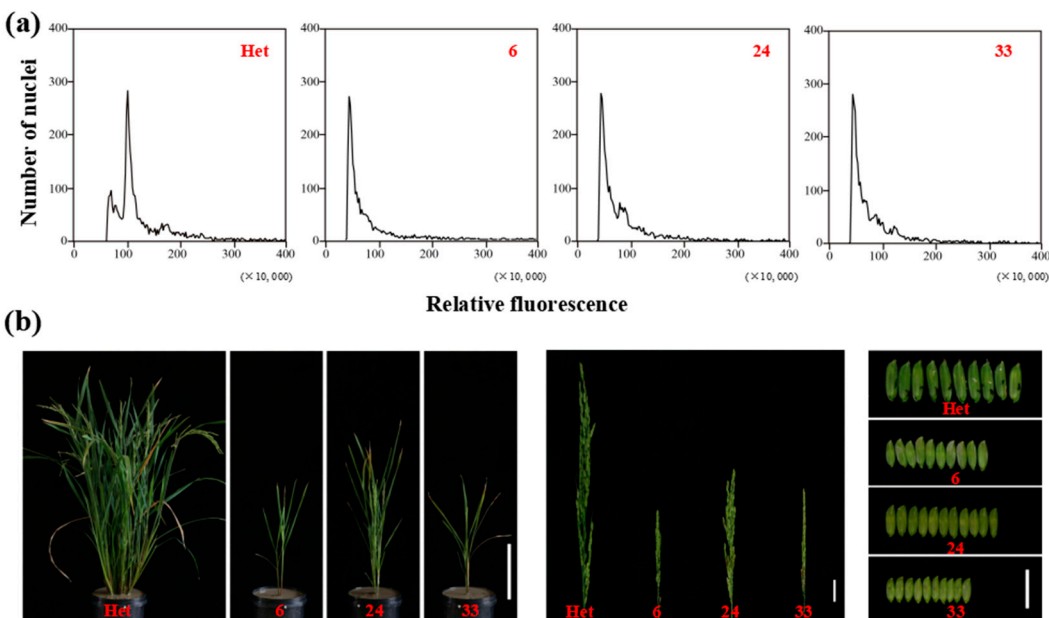

**Figure 8.** (**a**) Ploidy analysis of heterozygous diploid plant (Het) and haploid lines (6, 24, and 33) by flow cytometry. (**b**) Whole plants, panicles. and glumes of heterozygous diploid plant (Het) and haploid lines (6, 24, and 33). Bars = 20 cm (**left**), 3 cm (**middle**), and 2 cm (**right**).

## 4. Discussion

DH technology, based on an in vivo haploid inducer, is an efficient method for inbred development and has gained widespread use in modern maize breeding programs [2]. The identification of *MTL*, a conserved genetic factor in cereals for triggering haploid induction in maize, has facilitated the development of in vivo DH technology and synthetic apomixis systems in rice [4–6,8–10,17,20]. However, it is challenging to develop a high-throughput DH technology platform in rice.

DH technology encompasses induction, identification, and chromosome doubling of haploids [14]. Knockout of *OsMTL* produces 2–12% haploids in rice [8,9], which solves the problem of haploid induction. However, the step of haploid identification remains a limiting factor. Currently, there are four common methods available for identifying haploids in rice: (1) ploidy analysis using flow cytometry, (2) phenotypic comparison at the late stage of plant growth, (3) genotyping using molecular markers, and (4) phenotypic screening using visually discernible traits. Identification of haploids via methods (3) and (4) is more efficient and cost-effective compared to the time-consuming and expensive methods (1) and (2). In this study, we optimized methods (3) and (4) for haploid identification.

Molecular markers have long been deployed, in combination with flow cytometry, to determine homozygosity and identify haploids [21]. However, this method presents certain challenges due to the variability of female germplasms in the development of unique and universal InDel markers. To overcome this, we introduced large insertion or deletion mutations in the *MTL* gene, which allowed us to easily develop a unique and usable InDel marker without considering the variability in female germplasms. Our strategy can serve as an alternative method for haploid identification in maize, wheat, millet, and other species, as editing of the *MTL* gene has been shown to induce haploids in these species [4–6,22,23]. Furthermore, our strategy can also be employed in haploid identification for recently identified haploid genes such as *ZmPLD3* [24], *ZmPOD65* [25], and *DMP* [7]. Additionally, the identification of haploids through the use of unique InDel markers can be continuously optimized. With the development of various rapid and cost-effective DNA extraction techniques in plants [26,27], it will become more convenient and efficient to identify haploids by utilizing these methods in combination with the InDel markers.

Phenotypic marker systems have been widely used for screening haploids and greatly improve the efficiency of haploid identification in maize [1]. However, a visually discernible trait cannot always be found in rice. First, traits used as phenotypic markers for haploid identification must be dominant and visible at seed or early seedling stages, such as anthocyanin, high oil content, and red root markers [11–14]. Second, unlike big maize seeds, rice seeds are small and have husks. Fortunately, we identified hairy leaf, which was shown to be a single dominant trait that could be visible at the seedling stage. We integrated this trait into haploid inducers and found that it greatly improved the efficiency of haploid identification. As a result, we have made the first report of a phenotypic marker for identifying haploids in rice. In future, as the use of phenotypic markers greatly improves the efficiency of haploid identification, more markers such as anthocyanin can be integrated into haploid inducers. Furthermore, it is worth exploring the combination of different markers to enhance both the efficiency and accuracy of haploid identification.

### 5. Conclusions

We have successfully created haploid inducers in rice by introducing insertion/deletion mutations into the *OsMTL* gene and utilizing the hairy leaf trait, both of which serve as specific molecular and phenotypic markers. Our marker-based haploid inducers significantly enhance the efficiency of haploid identification, thus greatly accelerating the application of in vivo DH technology in rice breeding.

**Supplementary Materials:** The following supporting information can be downloaded at: https://www.mdpi.com/article/10.3390/agronomy13061520/s1. Table S1: The primers used in this study.

**Author Contributions:** C.L. and W.L. managed the project. J.W., H.Y. and J.R. performed the experiments. X.J., H.L. and F.H. analyzed the data. J.W. wrote the manuscript. C.L. revised the manuscript. All authors have read and agreed to the published version of the manuscript.

**Funding:** This work was supported by the National Natural Science Foundation of China (32001527) and the earmarked fund for CARS–01.

**Data Availability Statement:** All datasets used in this study are included in the manuscript and the supplementary file.

**Conflicts of Interest:** The authors declare that the research was conducted in the absence of any commercial or financial relationships that could be construed as a potential conflict of interest.

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
