# Peer review of "Development of Specific Molecular and Phenotypic Marker-Based Haploid Inducers in Rice"

_agronomy, doi:10.3390/agronomy13061520_

Round 1
Reviewer 1 Report
The MS titled "Development of Specific Molecular and Phenotypic Marker-based Haploid Inducers in Rice" addresses the critical issue of producing haploid plants in rice breeding programs. The authors present a comprehensive study that focuses on the development of marker-based haploid inducers using molecular and phenotypic markers.
Only minor English editing required of some sentence to maintained its verbatim, and second some references are too old, needs to update recent citation.
some sentence needs rephrasing
Reviewer 2 Report
The work of Wang et al. use a described application of the CRISPR/Cas9 methodology to induce the generation of haploid individuals and generating traceable genetic markers. Nevertheless, the method is known and authors must consider citing this article (https://link.springer.com/protocol/10.1007/978-1-0716-1068-8_14) or discuss their results comparing them with the approach used in the aforementioned document.
Reviewer 3 Report
There are some details that need to be attended to in the manuscript. The discussion should be improved by giving more weight to the discussion of the discovery or scientific contribution.

Reviewer 4 Report
DH technology, based on in vivo haploid inducer, is known to be an efficient strategy for inbred development and it is widely used in innovative maize breeding programs. The identification of MTL for triggering haploid induction in maize has facilitated the development of this technology. However, the haploid identification is still a limiting factor for the application of DH technology in other crops, such as rice, for which no marker-based haploid inducers have yet been developed.
In this work, the adopted strategy, based on the use of the CRISPR/Cas9 technology to create large InDel mutations on the OsMTL gene, seems to provide unique molecular markers that can be integrated into haploid inducers to aid in haploid identification. Consequently, these results could accelerate the application of in vivo DH technology in rice breeding.
Therefore, the adopted approach could represent a relevant improvement, compared to the existing state of the art in cereals, with particular attention to rice.
Material and methods are well described, while the Discussion section could be extended with major information about the impacts of the achieved results in rice breeding programs.
Minor editing of English language required
